# ASSISTED LEARNING FOR ORGANIZATIONS WITH LIMITED IMBALANCED DATA

## ABSTRACT

We develop an assisted learning framework for assisting organization-level learners to improve their learning performance with limited and imbalanced data. In particular, learners at the organization level usually have sufficient computation resource, but are subject to stringent data sharing and collaboration policies. Their limited imbalanced data often cause biased inference and sub-optimal decision-making. In our assisted learning framework, an organizational learner purchases assistance service from a service provider and aims to enhance its model performance within a few assistance rounds. We develop effective stochastic training algorithms for assisted deep learning and assisted reinforcement learning. Different from existing distributed algorithms that need to frequently transmit gradients or models, our framework allows the learner to only occasionally share information with the service provider, and still achieve a near-oracle model as if all the data were centralized.

## 1 INTRODUCTION

Modern distributed learning frameworks such as federated learning (Shokri & Shmatikov, 2015; Konecny et al., 2016; McMahan et al., 2017) aim to improve the learning performance for a large number of learners that have limited data and computation/communication resources. These learning frameworks are well suited for cloud systems and IoT systems (Ray, 2016; Gomathi et al., 2018) that manage numerous smart devices through wireless communication.

Over the past decade, many organizations, e.g., government agencies, hospitals, schools, and companies, have integrated machine learning models into their work pipeline to facilitate data analysis and decision making. For example, according to a recent survey (Financesonline, 2021), 49% of companies worldwide are exploring or planning to use machine learning, 51% of organizations claim to be early adopters of machine learning, and the estimated productivity improvement from the learning models is 40%. However, the performance of their machine learning models critically depends on the quality of the data, which typically is of a limited population and is biased toward certain distributions. Unfortunately, the existing learning frameworks cannot help big organizations improve their learning performance due to the following major restrictions.

- Unlike smart devices in the conventional federated learning, organizational learners typically cooperate with a single external service provider under a rigorous contract. Moreover, the service provider is often presumed to have more data with better quality than the organization.
- Conventional distributed learners achieve the performance goal by frequently exchanging information with other learners. In comparison, each learning round for organizational learners is costly, as they need to pay the provider for assistance and need to exchange a large amount of information with the provider. Hence, organizational learners desire to achieve a significant performance improvement within limited assistance rounds.

Therefore, there is an emerging need to develop a modern machine learning framework for organizational learners that can significantly improve the model performance by purchasing limited assistance services from external providers without data sharing. This constitutes the goal of this work.

In this work, we develop an assisted learning framework in the 'horizontal-splitting' setting, where the learner and the service provider possess different datasets that are utilized for training a common model. In our context, the learner's data is assumed to be limited and imbalanced, while the provider's

data is supposed to be big and complements the learner's data. Our learning framework nicely suits the organizational learners' characteristics: they have a very limited budget for purchasing external assistance services, yet they exchange a large amount of side information with the provider per assistance round to maximize the performance gain. This is opposite to federated learning, where smart devices are equipped with only a limited communication budget but can endlessly learn through interacting with the cloud. We summarize our contributions as follows.

## 1.1 OUR CONTRIBUTIONS

We identify the need for developing an assisted learning framework for facilitating the deployment of general machine learning in large organizations. This learning framework addresses the unique challenges as explained previously.

We first develop an *assisted deep learning* framework for organizational learners with limited and imbalanced data, and propose a stochastic training algorithm named AssistSGD. Specifically, every assistance round of AssistSGD consists of two phases. In the first phase, the learner performs local SGD training for multiple iterations and sends the generated trajectory of models together with their corresponding local loss values to the service provider. In the second phase, the provider utilizes the learner's information to evaluate the global loss of the received models, and uses the best model with the smallest global loss as an initialization. Then, the provider performs local SGD training for multiple iterations and sends the generated trajectory of models together with their corresponding local loss values to the learner. Finally, the learner utilizes the provider's information to evaluate the global loss of the received models, and outputs the best model with the smallest global loss. Under mild technical assumptions, we formally prove that AssistSGD with full batch gradient updates is guaranteed to find a critical point of the global loss function in general nonconvex optimization.

We further generalize the framework to enable *assisted reinforcement learning*, and develop a policy gradient training algorithm named AssistPG, which has the same training logic as that of AssistSGD.

Through extensive experiments with deep learning and reinforcement learning, we demonstrate that the learner can achieve a near-oracle performance with AssistSGD and AssistPG as if all the data were centralized. In particular, as the learner data's level of imbalance increases, AssistSGD can help the learner achieve a higher performance gain. Moreover, data are never exchanged in the assisted learning process for both participants.

## 1.2 RELATED WORK

**Assisted learning**. Earlier work on assisted learning (Xian et al., 2020) considers organizations that collect different features from the same cohort. This is in contrast with our context where organizations hold the same features but imbalanced data distributions or environments. Also, our method applies to general deep learning and reinforcement learning tasks, which are beyond the previously studied regression task. Consequently, our algorithm designs and application scenarios are significantly different from the prior work.

**Distributed optimization**. In conventional distributed optimization, the data is evenly distributed among workers, which collaboratively solve a large-scale problem by exchanging local information (gradients, models.) via either decentralized networks (Xie et al., 2016; Lian et al., 2017; 2018) or centralized networks (Ho et al., 2013; Li et al., 2014; Richtarik & Takavc, 2016; Zhou et al., 2016; 2018). As a comparison, our AssistSGD only requires a few transmission rounds between the learner and provider. This is particularly appealing for organizational learners, who can employ a sophisticated optimization process locally while restricting the rounds of assistance.

**Federated learning**. Federated learning is an emerging distributed learning framework (Shokri & Shmatikov, 2015; Konecny et al., 2016; McMahan et al., 2017; Zhao et al., 2018; Li et al., 2020; Diao et al., 2021), which aims to learn a global model using the average of local models trained by numerous smart devices with heterogeneous data. The existing federated learning algorithms require frequent transmissions of local model parameters. This is different from our solution designed for the organizational learning scenarios, where each learner is an organization that often has unconstrained communication and computation resources, but is restricted to interact with other external service providers. Our solution aims to help the learner improve learning performance within ten rounds, while federated learning needs many more rounds.

## 2 ASSISTED DEEP LEARNING

In this section, we introduce the assisted deep learning framework. Throughout the paper, L denotes a learner who seeks assistance, and P denotes a service provider who provides assistance to L.

### 2.1 PROBLEM FORMULATION

We consider the case where the learner L aims to train a machine learning model $\theta$ that performs well on its own dataset $\mathcal{D}^{(L)}$ and generalizes well to unseen data. In general, L can train a machine learning model by solving the empirical risk minimization problem $\min_{\theta \in \Theta} f(\theta; \mathcal{D}^{(L)})$, where $f(\cdot; \mathcal{D}^{(L)})$ is the loss on $\mathcal{D}^{(L)}$ and $\Theta$ is the parameter space. Standard statistical learning theories show that the obtained model can generalize well to intact test samples under suitable constraints of model parsimoniousness (Ding et al., 2018). However, when the user's data $\mathcal{D}^{(L)}$ contains a **limited** number of samples that are highly **imbalanced**, the learned model will suffer from overfitting or deteriorated generalization capability to the unseen test data.

To overcome data deficiency, the learner L intends to connect with an external service provider P (e.g., a commercialized data company), who possesses data $\mathcal{D}^{(P)}$ that are sufficient or complementary to the learner's data $\mathcal{D}^{(L)}$. Ideally, the user L would improve the model by solving the following data-augmented problem, where $\mathcal{D}^{(L,P)} := \mathcal{D}^{(L)} \cup \mathcal{D}^{(P)}$ denotes the centralized data.

$$\theta^{(L,P)} = \underset{\theta \in \Theta}{\arg\min} \, f(\theta; \mathcal{D}^{(L,P)}), \quad \text{where } \mathcal{D}^{(L,P)} = \mathcal{D}^{(L)} \cup \mathcal{D}^{(P)}. \tag{1}$$

We note that $f(\theta; \mathcal{D}^{(L,P)}) = f(\theta, \mathcal{D}^{(L)}) + f(\theta, \mathcal{D}^{(P)})$. If $\mathcal{D}^{(P)}$ is generated from a distribution that is close to the underlying data distribution, then it is expected that $\theta^{(L,P)}$ will achieve significantly improved performance on unseen data. However, it is unrealistic to centralize the data since the interactions between the learner L and the provider P are often restricted by various regulations. Some representative regulations that formally define the assisted learning framework are listed below.

---

**Assisted Learning Protocols**

1. *No data sharing:* Neither the learner L nor the provider P will share data with each other.

2. *Limited assistance:* The learner L has a limited budget for purchasing assistance service. The learner desires to maximize the performance gain with only a few assistance rounds.

3. *Unlimited communication bandwidth:* In each assistance round, the learner and the provider can exchange unlimited information. For example, the learner (resp. provider) can send an employee (resp. technician) to deliver a large-capacity hard drive to the other.

---

The above assisted learning framework is different from the existing learning frameworks. For example, in federated learning, many devices collaboratively train a global model via a large number of learning rounds with limited communication bandwidth. In comparison, the organizational learner in assisted learning can only query a few rounds of assistance from the provider, but can exchange unlimited information with it.

Hence, we need to develop a training algorithm that can substantially improve the learner's model quality using limited interactions with the provider for assisted learning. Next, we present an assisted stochastic gradient descent (AssistSGD) algorithm for this purpose.

### 2.2 ASSISTSGD FOR ASSISTED DEEP LEARNING

We propose *AssistSGD* in Algorithm 1 for assisted deep learning. The learning process consists of $R$ rounds, each consisting of the following interactions between the learner L and the provider P.

(1) First, the learner L initiates a local learning process. It initializes a model $\theta_0^{(L)}$ and applies SGD with learning rate $\eta$ to update it for $T$ iterations using the local dataset $\mathcal{D}^{(L)}$. Then, the learner evaluates the local loss $f(\cdot; \mathcal{D}^{(L)})$ in a subset $\mathcal{T}$ of the iterations $t = 0, 1, ...T-1$. Lastly, the learner sends this subset of models and their corresponding local loss to the provider P .

(2) Upon receiving the information from the learner L, the provider P first evaluates the global loss $f(\cdot; \mathcal{D}^{(\mathrm{L,P})})$ of the received set of models $\{\theta_t^{(\mathrm{L})}, t \in \mathcal{T}\}$ and picks the best one (denoted by $\theta_0^{(\mathrm{P})}$) for initialization. Note that the global loss can be evaluated because the local loss $\{f(\theta_t^{(\mathrm{L})}; \mathcal{D}^{(\mathrm{L})}), t \in \mathcal{T}\}$ are provided by the learner L , and the provider P just needs to evaluate the local loss $\{f(\theta_t^{(\mathrm{L})}; \mathcal{D}^{(\mathrm{P})}), t \in \mathcal{T}\}$. After that, the provider applies SGD with learning rate $\eta$ to update the model for $T'$ iterations on the local dataset $\mathcal{D}^{(\mathrm{P})}$. Then, the provider evaluates the local loss $f(\cdot; \mathcal{D}^{(\mathrm{P})})$ in a subset $\mathcal{T}'$ of the iterations $t = 0, 1, ...T' - 1$, and sends the subset of models and their corresponding local loss to the learner L.

---

**Algorithm 1** AssistSGD

**Input:** Initialization model $\theta^0$, learning rate $\eta$, assistance rounds $R$, local iterations $T$.
**for** assistance rounds $r = 1, \ldots, R$ **do**

> **Learner L :**
> ▶ Initialize $\theta_0^{(\mathrm{L})} = \theta^{r-1}$.
> ▶ Local SGD training to generate $\{\theta_t^{(\mathrm{L})}\}_{t=0}^{T-1}$.
> ▶ Send $\{\theta_t^{(\mathrm{L})}, f(\theta_t^{(\mathrm{L})}; \mathcal{D}^{(\mathrm{L})})\}_{t \in \mathcal{T}}$ to provider P .
>
> ---
>
> **Provider P :**
> ▶ Initialize $\theta_0^{(\mathrm{P})} = \arg\min_{\theta \in \{\theta_t^{(\mathrm{L})}\}_{t \in \mathcal{T}}} f(\theta; \mathcal{D}^{(\mathrm{L,P})})$.
> ▶ Local SGD training to generate $\{\theta_t^{(\mathrm{P})}\}_{t=0}^{T'-1}$.
> ▶ Send $\{\theta_t^{(\mathrm{P})}, f(\theta_t^{(\mathrm{P})}; \mathcal{D}^{(\mathrm{P})})\}_{t \in \mathcal{T}'}$ to learner L .
>
> ---
>
> **Learner L :**
> ▶ Output $\theta^r = \arg\min_{\theta \in \{\theta_t^{(\mathrm{P})}\}_{t \in \mathcal{T}'}} f(\theta; \mathcal{D}^{(\mathrm{L,P})})$.

**end**
**Output:** The best model in $\{\theta^r\}_{r=1}^R$.

---

(3) Once the learner L receives feedback from the provider P, it evaluates the global loss $f(\cdot; \mathcal{D}^{(\mathrm{L,P})})$ of the received set of models $\{\theta_t^{(\mathrm{P})}, t \in \mathcal{T}'\}$ and picks the best model as the output model of this assistance round.

**Discussions**. The above algorithm works for general deep learning tasks. It does not require data sharing between the learner and the provider. Moreover, in each learning round, the learner and the provider exchange a small number of their local training models. As we show later in the experimental studies, it suffices to sample the iterations in $\mathcal{T}, \mathcal{T}'$ at a low frequency. Such an assisted learning process is very different from, for example, the federated learning process. Particularly, in each round of federated learning, all learners perform a small number of local SGD updates and send their last output models to the cloud due to limited communication bandwidths. Consequently, the global model needs a large number of learning rounds to achieve a desirable performance.

## 2.3 Convergence Analysis

In this subsection, we show that AssistSGD provably converges to a stationary point in smooth nonconvex optimization. For simplicity, we consider the deterministic setting, where the local training of AssistSGD uses full gradient updates. We also make the following reasonable assumptions.

**Assumption 1.** *We assume that the assisted learning problem in equation 1 satisfies the following conditions.*

1. *The global loss $f(\theta; \mathcal{D}^{(\mathrm{L,P})})$ has L-Lipschitz gradients. Moreover, $\inf_\theta f(\theta; \mathcal{D}^{(\mathrm{L,P})}) > -\infty$;*

2. *There exists a $G > 0$ such that $\max\{\|\nabla f(\theta; \mathcal{D}^{(\mathrm{L})})\|, \|\nabla f(\theta; \mathcal{D}^{(\mathrm{P})})\|, \|\nabla f(\theta; \mathcal{D}^{(\mathrm{L,P})})\|\} \le G$ for all $\theta$ generated by AssistSGD. This holds when the generated model trajectory is bounded.*

Note that in each assistance round $r$, both the provider and the learner pick the best model via the $\arg\min$ operation to initialize and finalize their local training. This guarantees that AssistSGD continuously makes optimization progress. Specifically, we let $\theta_0^{(\mathrm{L}),r}, \theta_0^{(\mathrm{P}),r}$ denote learner L's and provider P's initialization models, respectively, in the round $r$. Recall that $\theta^r$ is the output model of learner L (see Algorithm 1). Then, the two $\arg\min$ operations guarantee that the following proposition holds.

**Proposition 1.** *The sequence of global loss $\{f(\theta^r; \mathcal{D}^{(\mathrm{L,P})})\}_r$ achieved by AssistSGD monotonically decreases to a finite limit, namely, in any round $r$,*

$$f(\theta^r, \mathcal{D}^{(\mathrm{L,P})}) \le f(\theta_0^{(\mathrm{P}),r}, \mathcal{D}^{(\mathrm{L,P})}) \le f(\theta_0^{(\mathrm{L}),r}, \mathcal{D}^{(\mathrm{L,P})}) = f(\theta^{r-1}, \mathcal{D}^{(\mathrm{L,P})}).$$

Next, we prove that the output model $\theta^r$ asymptotically converges to a stationary point.

**Theorem 1.** *Let Assumption 1 hold and run AssistSGD with full gradient updates for $R$ assistance rounds. With $\eta = \mathcal{O}((RLTG^2)^{-0.5})$, we have that $\min_{0 \le r \le R-1} \|\nabla f_{\mathrm{L,P}}(\theta^r)\| \to 0$ as $R \to \infty$.*

Therefore, with a proper choice of the learning rate $\eta$, assisted learning is guaranteed to find a stationary point in general nonconvex optimization. We note that it is generally hard to establish a tight convergence complexity bound for AssistSGD due to the uncertainty of the $\arg\min$ operations. Still, our experiments show that it can often achieve the performance of centralized SGD training.

## 3 ASSISTED REINFORCEMENT LEARNING

In this section, we extend our assisted learning framework to Reinforcement Learning (RL) scenarios to enhance the model generalizability. We first introduce some basic setup of RL.

**Markov Decision Process (MDP).** We consider a standard finite-horizon MDP that is denoted by a tuple $M = (\mathcal{S}, \mathcal{A}, \mathbf{P}, r, \pi, \rho_0, T)$, where $\mathcal{S}$ is the state space, $\mathcal{A}$ corresponds to the action space, $\mathbf{P} : \mathcal{S} \times \mathcal{A} \times \mathcal{S} \to [0, 1]$ denotes the underlying state transition kernel that drives the new state given the previous state and action, $r : \mathcal{S} \times \mathcal{A} \mapsto \mathbb{R}$ is the reward function, $\pi : \mathcal{S} \to \mathcal{A}$ is the policy, $\rho_0$ denotes the initial state distribution, and $T$ is the episode length. Given a policy $\pi_\theta$ parameterized with $\theta$, the goal of RL, also known as on-policy learning, is to learn an optimal policy parameter $\theta^*$ that maximizes the expected accumulated reward, namely $\theta^* = \arg\max_\theta J(\theta) := \mathbb{E}[\sum_{t=1}^{T} \gamma^{t-1} r_t]$, where the expectation is taken with respect to the finite-length episode.

### 3.1 PROBLEM FORMULATION

We assume that an RL learner L has collected a small amount of Markovian data $\mathcal{D}^{(\mathrm{L})}$ by interacting with a certain environment. It wants to train a policy that generalizes well to other similar environments. However, the data and environment that the learner L can access are limited. In assisted reinforcement learning, the learner L aims to enhance its policy's generalizability to unseen environments by querying assistance from a service provider P. For example, autonomous-driving startup companies typically own limited data that are insufficient for training good autonomous driving models that perform well in heterogeneous environments, and they can purchase assistance service from big companies (who own massive data) to improve the model performance and generalizability.

Formally, we assume that there is an underlying distribution of transition kernel that models the variability of the environment. Specifically, denote $E_\beta$ as an environment with the transition kernel $\mathbf{P}_\beta$ parameterized by $\beta$, which follows an underlying distribution $q$. Let $J_\beta(\theta)$ denote the expected accumulated reward collected from the environment $E_\beta$ following the policy $\pi_\theta$. The learner L's ultimate goal is to learn a good policy that applies to the underlying distribution of environment, namely, $\max_\theta \mathbb{E}_{\beta \sim q}\big[J_\beta(\theta)\big]$. In practice, the learner L only has training data collected from a limited number of environment instances, say $\beta^{(\mathrm{L})} = \{\beta_1^{(\mathrm{L})}, \ldots, \beta_{n_{\mathrm{L}}}^{(\mathrm{L})}\}$. On the other hand, the service provider may have rich experience interacting with a more diverse set of environments, say $\beta^{(\mathrm{P})} = \{\beta_1^{(\mathrm{P})}, \ldots, \beta_{n_{\mathrm{P}}}^{(\mathrm{P})}\}$. Consequently, the learner aims to solve the following assisted RL problem.

$$\max_\theta J_{\beta^{(\mathrm{L,P})}}(\theta) := \sum_{\beta \in \beta^{(\mathrm{L})}} J_\beta(\theta) + \sum_{\beta' \in \beta^{(\mathrm{P})}} J_{\beta'}(\theta) \tag{2}$$

which is similar to the formulation of assisted deep learning. In the following subsection, we will develop a policy gradient (PG)-type algorithm for solving the assisted RL problem in equation 2.

### 3.2 ASSISTPG FOR ASSISTED REINFORCEMENT LEARNING

Policy gradient (PG) is a classic RL algorithm for policy optimization. The PG algorithm estimates the policy gradient $\nabla J(\theta)$ via the policy gradient theorem, and applies it to update the policy. Specifically, given one episode $\tau$ with length $T$ that is collected under the current policy $\pi_\theta$, the corresponding policy gradient takes the following form, where $R(\tau) = \sum_{t=1}^{T} \gamma^{t-1} r_t$ is the discounted accumulated

reward over this episode. In practice, a mini batch of episodes are used to estimate the policy gradient.

$$\nabla J(\theta) \approx R(\tau) \sum_{t=1}^{T} \nabla \log \pi_\theta\big(a_t^{(i)}|s_t^{(i)}\big)$$

In Algorithm 2, we present **Assisted Policy Gradient (AssistPG)**–a policy gradient-type algorithm for solving the assisted RL problem in equation 2. The main logic of the AssistPG algorithm is the same as that of the AssistSGD for assisted deep learning.

---

**Algorithm 2** AssistPG

**Input:** Initialization model $\theta^0$, learning rate $\eta$, assistance rounds $R$, local iterations $T$.

**for** assistance rounds $r = 1, \ldots, R$ **do**

    **Learner L :**
        ▶ Initialize $\theta_0^{(L)} = \theta^{r-1}$.
        ▶ Local PG training to generate $\{\theta_t^{(L)}\}_{t=0}^{T-1}$.
        ▶ Send $\{\theta_t^{(L)}, \sum_{\beta \in \beta^{(L)}} J_\beta(\theta_t^{(L)})\}_{t \in \mathcal{T}}$ to provider P .

    **Provider P :**
        ▶ Initialize $\theta_0^{(P)} = \arg\max_{\theta \in \{\theta_t^{(L)}\}_{t \in \mathcal{T}}} J_{\beta^{(L,P)}}(\theta)$.
        ▶ Local PG training to generate $\{\theta_t^{(P)}\}_{t=0}^{T'-1}$.
        ▶ Send $\{\theta_t^{(P)}, \sum_{\beta \in \beta^{(P)}} J_\beta(\theta_t^{(P)})\}_{t \in \mathcal{T}'}$ to learner L .

    **Learner L :**
        ▶ Output $\theta^r = \arg\max_{\theta \in \{\theta_t^{(P)}\}_{t \in \mathcal{T}'}} J_{\beta^{(L,P)}}(\theta)$.

**end**

**Output:** The best model in $\{\theta^r\}_{r=1}^{R}$.

---

## 4 EXPERIMENTS

In this section, we first visualize AssistSGD training to help understand the mechanism of assisted learning. Then, we provide extensive experiments of deep learning and reinforcement learning to demonstrate the effectiveness of the proposed assisted learning algorithms.

### 4.1 VISUALIZATION OF ASSISTSGD TRAINING

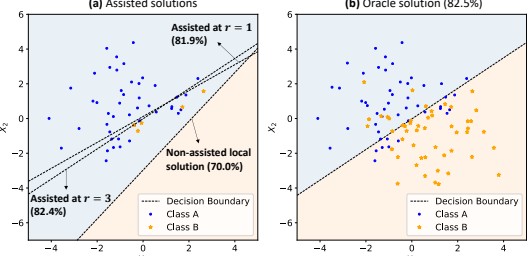

Figure 1: Visualization of AssistSGD: (a) the learner's classifiers after being assisted by the provider at different rounds, and (b) oracle classifier obtained by using centralized data. The test accuracies are shown in the parentheses.

We visualize the learning process of AssistSGD through a simple logistic regression task. Consider two classes of data samples: Class A data contains $50$ points drawn from $\mathcal{N}([-1, 1], 1.5^2 I_2)$ , and Class B data contains $50$ points drawn from $\mathcal{N}([1, -1], 1.5^2 I_2)$, where $\mathcal{N}$ and $I$ denote Gaussian distribution and identity matrix, respectively. Suppose that a learner L observes 90% class A samples and 10% class B samples. Another service provider P observes a similar number of data samples consisting of 10% class A samples and 90% class B samples. The learning process of AssisSGD is illustrated in Figure 1 (Left), and we also show the oracle solution trained by SGD with centralized data in Figure 1 (Right).

It can be seen that without any assistance, the learner can only achieve 70% accuracy and the corresponding classifier performs poorly on the samples in class B. In comparison, after a single round of assistance, the classification performance is significantly improved to 81.9% accuracy. After

three rounds of assistance, the corresponding classifier is relatively close to the oracle classifier, which achieves an accuracy of 82.5%. Hence, it can be seen that AssistSGD has the potential to achieve a near-oracle performance. We also present the visualization of another regression example in Section A.2 of the Appendix.

## 4.2 Assisted Deep Learning Experiments

We test the performance of AssistSGD by comparing it with three baselines: standard SGD (using centralized data $\mathcal{D}^{(L,P)}$), Learner-SGD (using only the learner's data $\mathcal{D}^{(L)}$), and the FedAvg algorithm (McMahan et al., 2017) for federated learning. We implement these algorithms to train an AlexNet (with learning rate 0.01 and batch size 256) and a ResNet-18 (with learning rate 0.1 and batch size 256) on the CIFAR-10 dataset (Krizhevsky, 2009).

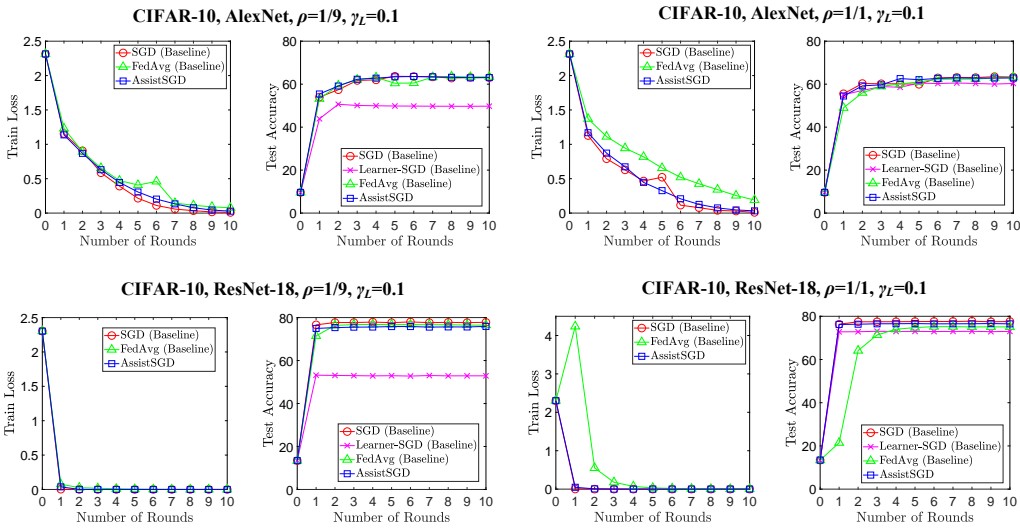

Figure 2: Comparison of AssistSGD, SGD, Learner-SGD and FedAvg with balanced learner's data using AlexNet (top row) and ResNet-18 (bottom row).

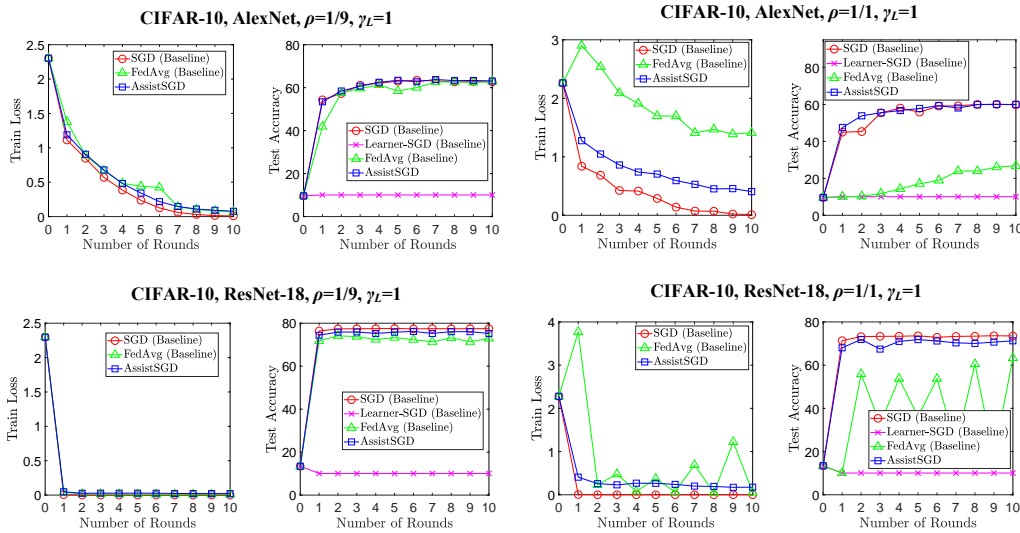

Figure 3: Comparison of AssistSGD, SGD, Learner-SGD and FedAvg with imbalanced learner's data using AlexNet (top row) and ResNet-18 (bottom row).

For AssistSGD, we distribute the entire training set of CIFAR-10 (50k samples) to the learner and provider according to two parameters: 1) data size ratio $\rho := |\mathcal{D}^{(L)}|/|\mathcal{D}^{(P)}|$, and 2) data imbalance ratio $\gamma_L \in (0, 1)$ that specifies the imbalance level of the learner's data. Specifically, we first randomly

assign one sample class as the primary class of the learner's data. Then, $\gamma_L \cdot |\mathcal{D}^{(L)}|$ number of samples are sampled from the primary class, and the rest $(1 - \gamma_L) \cdot |\mathcal{D}^{(L)}|$ number of samples are drawn from the remaining classes uniformly at random. The provider's data are sampled from all classes uniformly at random and are balanced. We also fix the number of assistance rounds to be 10. The total number of local SGD iterations in each assistance round is fixed to be 2000. We assign the SGD iteration budget to the learner and provider in proportion to their data samples' sizes. Both the learner and provider record their local training models and local loss values for every $I = 50$ SGD iterations, which is the sampling period. For FedAvg, we treat the learner and provider as two federated learning agents, and they inherit the same local data and local SGD iteration budgets from AssistSGD.

We first compare these algorithms with balanced learner's data $\gamma_L = 0.1$ and varying data size ratios $\rho = 1/9, 1/1$. We plot the training loss (on centralized data $\mathcal{D}^{(L,P)}$) and the test accuracy (on the 10k test data) against the number of assistance rounds in Figure 2 (top row AlexNet, bottom row ResNet-18). Here, one assistance round on the x-axis is interpreted as 2000 SGD iterations for SGD, Learner-SGD and FedAvg. The training loss of Learner-SGD is not reported as it is trained on $\mathcal{D}^{(L)}$ only. It can be seen that AssistSGD achieves a comparable performance to that of SGD with centralized data. In particular, when $\rho = 1/9$ and the learner has limited data, the test performance of AssistSGD is significantly better than Learner-SGD, demonstrating the effectiveness of querying assistance from the service provider. When $\rho = 1/1$ and the learner has more data, AssistSGD still achieves a near-oracle performance. In particular, it converges much faster and achieves a slightly better test performance than FedAvg.

We further test and compare these algorithms with imbalanced learner's data $\gamma_L = 1$ and $\rho = 1/9, 1/1$. The results are shown in Figure 3. It can be seen that when $\rho = 1/9$ and the learner has limited data, AssistSGD achieves a comparable performance to that of SGD, and slightly outperforms FedAvg. Moreover, when $\rho = 1/1$, AssistSGD converges slower than SGD due to the large amount of highly imbalanced data $\mathcal{D}^{(L)}$. Nevertheless, it still achieves a comparable test performance to that of SGD. On the other hand, AssistSGD significantly outperforms FedAvg, demonstrating the robustness of assisted learning to data heterogeneity. Comparing the results in Figure 3 with those in Figure 2, we conclude that AssistSGD improves the test performance more (compare to Learner-SGD and FedAvg) when the learner's data are more imbalanced.

Due to space limitation, we present other CIFAR-10 training results of both network models under the parameters $\rho = 1/3$ and $\gamma_L = 0.1, 1$ and the corresponding SVHN training results in Section A.3.1 of the Appendix, where one can make the same conclusions for both datasets. We also explore the effect of sampling period (Section A.3.2) on the performance of AssistSGD in the Appendix.

### 4.3 ASSISTED REINFORCEMENT LEARNING EXPERIMENTS

We demonstrate the effectiveness of AssistPG via solving two reinforcement learning problems: the CartPole (Barto et al., 1983) and the LunarLander provided by the OpenAI Gym library (Brockman et al., 2016). In the CartPole problem, a controller aims to stabilize a pole attached to a cart by applying left or right force to the cart (see the first figure in Figure 4), and we show that AssistPG can help the controller stabilize the pole with different pole lengths. For the LunarLander problem, a lander initializes its landing from top left of the sky and aims to land on a landing pad by controlling its engine (see the first figure in Figure 5). We show that AssistPG can help land the lander with different engine powers.

We assume that the learner and the provider can query episode data by interacting with diverse environments. Specifically, for the CartPole problem, we parameterize the environment using the pole length. Both the learner and the provider train their control policies by playing 5 Cartpole games with the pole length randomly generated from Uniform$(4, 5)$ (for the learner) and Uniform$(0, 1)$ (for the provider). For the LunarLander problem, we parameterize the environment using the engine power. Both the learner and provider train their control policies by playing 10 LunarLander games with the engine power randomly generated from Uniform$(10, 15)$ (for the learner) and Uniform$(35, 40)$ (for the provider). Moreover, we consider two sets of testing environments that are uniform ("Test I") and non-uniform ("Test II"), respectively. For the CartPole problem, Test I environments randomly generate the pole length from Uniform$(0, 5)$, and Test II environments randomly generate the pole length from Beta$(1, 5)$ with probability 0.2 and Uniform$(0, 5)$ with probability 0.8. For the LunarLander

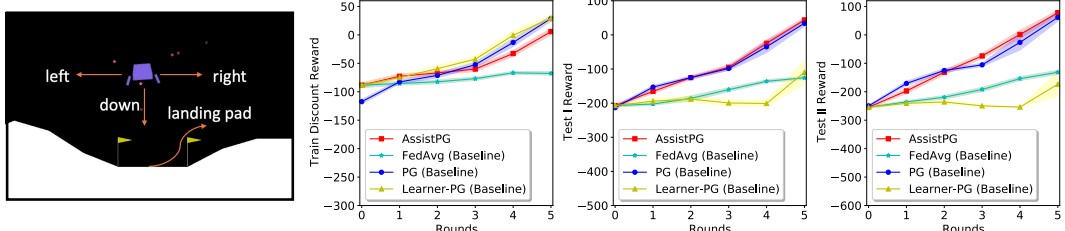

Figure 4: Comparison of AssistPG, PG, Learner-PG and FedAvg in the CartPole game.

Figure 5: Comparison of AssistPG, PG, Learner-PG and FedAvg in the LunarLander game.

problem, Test I environments randomly generate the engine power from Uniform$(10, 40)$, and Test II environments randomly generate the engine power as $30r + 10$, where $r \sim \text{Beta}(5, 1)$.

We test AssistPG on both RL problems and compare its performance with three baselines: the standard PG (using centralized episode data), the Learner-PG (using only learner's episode data), and the FedAvg algorithm that uses policy gradient updates. All these algorithms are implemented with learning rate $5 \times 10^{-3}$ and episode batch size 32. We model the policy using a three-layer feed-forward neural network with 4 and 32 hidden neurons for CartPole and LunarLander, respectively. Moreover, for AssistPG, we fix the total number of assistance rounds to be 10 and 5 for CartPole and LunarLander, respectively. The total number of local PG iterations in each assistance round is fixed to be 20 for both problems. We also set the sampling period to be four, namely, the learner and the provider record their local model and discounted training reward for every four local PG iterations. Figures 4 and 5 plot the discounted training rewards (collected in local environment only), Test I cumulative rewards, and Test II cumulative rewards against the assistance round obtained by all these algorithms, for solving the CartPole and LunarLander problems, respectively. Here, one assistant round is interpreted as 20 local PG iterations for algorithms other than AssistPG.

Figure 4 indicates that AssistPG outperforms Learner-PG and Fed-Avg, when the testing environments include diverse lengths of poles. Also, AssistPG can achieve a comparable performance to that of the PG with centralized data. Moreover, Figure 5 indicates that AssistPG can swiftly adapt to scenarios out of their comfort zone (namely the training environments) in only a few rounds. These experiments demonstrate that our assisted learning framework can help the learner significantly improve the quality of the policy for handling complex RL problems. Extensions and more details of these reinforcement learning experiments as well as video demonstrations are included in Section A.4 of the Appendix.

## 5  CONCLUSION

This work develops a new learning framework for assisting organizational learners to improve their learning performance with limited imbalanced data. In particular, the proposed AssistSGD and AssistPG allow the provider to assist the learner's training process and significantly improve its model quality within only a few assistance rounds. We demonstrate the effectiveness of both assisted learning algorithms through experimental studies. In the future, we expect that this learning framework can be integrated with other learning frameworks such as meta-learning and multi-task learning. A limitation of this study is that it only considers a pair of learner and provider. An interesting future direction is to emulate the current assisted learning framework to allow multiple learners or service providers.

The **Appendix document** contains the proof of the technical result, more experimental details, and video demonstrations of the reinforcement learning performance.

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

# A   APPENDIX

## A.1   PROOF OF THEOREM 1

For brevity, throughout the proof, we denote the loss functions $f(\cdot; \mathcal{D}^{(L)}), f(\cdot; \mathcal{D}^{(P)}), f(\cdot; \mathcal{D}^{(L,P)})$ as $f_L, f_P, f_{L,P}$, respectively.

We consider any assistance round $r$, and first study the local training of the learner L. Recall that the learner L initializes with the output model obtained from the previous round, namely $\theta_0^{(L),r} = \theta^{r-1}$. In the local training, learner L performs $T$ local gradient descent steps and generates the trajectory $\{\theta_t^{(L),r}\}_{t=0}^T$. Then, provider P picks the best model from this trajectory that achieves the minimum global loss, and we denote this model as $\theta_k^{(L),r}$ for certain $k \in \{0, ..., T\}$. It is clear that $f_{L,P}(\theta_k^{(L),r}) \le f_{L,P}(\theta_t^{(L),r})$ for all $t$. By smoothness of the global loss and the gradient descent update rule, we have that

$$f_{L,P}(\theta_{k+1}^{(L),r}) \le f_{L,P}(\theta_k^{(L),r}) + \langle \theta_{k+1}^{(L),r} - \theta_k^{(L),r}, \nabla f_{L,P}(\theta_k^{(L),r}) \rangle + \frac{L}{2}\|\theta_{k+1}^{(L),r} - \theta_k^{(L),r}\|^2$$

$$= f_{L,P}(\theta_k^{(L),r}) + \langle -\eta \nabla f_L(\theta_k^{(L),r}), \nabla f_{L,P}(\theta_k^{(L),r}) \rangle + \frac{L\eta^2}{2}\|\nabla f_L(\theta_k^{(L),r})\|^2.$$

Since $f_{L,P}(\theta_k^{(L),r}) \le f_{L,P}(\theta_{k+1}^{(L),r})$, the above inequality further implies that

$$\langle \nabla f_L(\theta_k^{(L),r}), \nabla f_{L,P}(\theta_k^{(L),r}) \rangle \le \frac{L\eta}{2}\|\nabla f_L(\theta_k^{(L),r})\|^2 \le \frac{LG^2\eta}{2}. \tag{3}$$

Next, we consider the local training of the provider P. Recall that the provider P will initialize with the best model sent by the learner, namely $\theta_0^{(P),r} = \theta_k^{(L),r}$. In the local training, the provider P performs $T'$ local gradient descent steps and generates the trajectory $\{\theta_t^{(P),r}\}_{t=0}^{T'}$. According to the smoothness of the global loss, we have that

$$f_{L,P}(\theta_1^{(P),r}) \le f_{L,P}(\theta_0^{(P),r}) + \langle \theta_1^{(P),r} - \theta_0^{(P),r}, \nabla f_{L,P}(\theta_0^{(P),r}) \rangle + \frac{L}{2}\|\theta_1^{(P),r} - \theta_0^{(P),r}\|^2$$

$$= f_{L,P}(\theta_0^{(P),r}) + \langle -\eta \nabla f_P(\theta_0^{(P),r}), \nabla f_{L,P}(\theta_0^{(P),r}) \rangle + \frac{L\eta^2}{2}\|\nabla f_P(\theta_0^{(P),r})\|^2$$

$$= f_{L,P}(\theta_0^{(P),r}) + \langle -\eta \nabla f_P(\theta_k^{(L),r}), \nabla f_{L,P}(\theta_k^{(L),r}) \rangle + \frac{L\eta^2}{2}\|\nabla f_P(\theta_0^{(P),r})\|^2$$

$$= f_{L,P}(\theta_0^{(P),r}) - \eta\big(\|\nabla f_{L,P}(\theta_k^{(L),r})\|^2 - \langle \nabla f_L(\theta_k^{(L),r}), \nabla f_{L,P}(\theta_k^{(L),r}) \rangle\big) + \frac{L\eta^2}{2}\|\nabla f_P(\theta_0^{(P),r})\|^2$$

$$\le f_{L,P}(\theta_0^{(P),r}) - \eta\|\nabla f_{L,P}(\theta_k^{(L),r})\|^2 + \frac{LG^2\eta^2}{2} + \frac{LG^2\eta^2}{2}, \tag{4}$$

where the last inequality utilizes equation 3 and the boundedness of the gradient. Denote $\theta_k^{(P),r}$ as the best model from this trajectory that achieves the minimum global loss. The above Inequality (4) and Proposition 1 further imply that

$$f_{L,P}(\theta^r) = f_{L,P}(\theta_{k'}^{(P),r}) \quad \text{(for some } k' \in \{0, \ldots, T'\})$$

$$\le f_{L,P}(\theta_1^{(P),r})$$

$$\le f_{L,P}(\theta_0^{(P),r}) - \eta\|\nabla f_{L,P}(\theta_k^{(L),r})\|^2 + LG^2\eta^2$$

$$\le f_{L,P}(\theta^{r-1}) - \eta\|\nabla f_{L,P}(\theta_k^{(L),r})\|^2 + LG^2\eta^2$$

$$\le f_{L,P}(\theta^{r-1}) - \eta\|\nabla f_{L,P}(\theta_0^{(L),r})\|^2 - \eta\|\nabla f_{L,P}(\theta_k^{(L),r}) - \nabla f_{L,P}(\theta_0^{(L),r})\|^2 +$$

$$+ 2\eta\|\nabla f_{L,P}(\theta_k^{(L),r}) - \nabla f_{L,P}(\theta_0^{(L),r})\|\|\nabla f_{L,P}(\theta_0^{(L),r})\| + LG^2\eta^2$$

$$\overset{(i)}{\le} f_{L,P}(\theta^{r-1}) - \eta\|\nabla f_{L,P}(\theta_0^{(L),r})\|^2 + 2\eta^2 LTG^2 + LG^2\eta^2$$

$$\le f_{L,P}(\theta^{r-1}) - \eta\|\nabla f_{L,P}(\theta^{r-1})\|^2 + 3\eta^2 LTG^2, \tag{5}$$

where the inequality (i) uses the fact that

$$
\begin{aligned}
2\eta\|\nabla f_{\mathrm{L,P}}(\theta_k^{(\mathrm{L}),r}) - \nabla f_{\mathrm{L,P}}(\theta_0^{(\mathrm{L}),r})\| \|\nabla f_{\mathrm{L,P}}(\theta_0^{(\mathrm{L}),r})\| &\leq 2\eta G\|\nabla f_{\mathrm{L,P}}(\theta_k^{(\mathrm{L}),r}) - \nabla f_{\mathrm{L,P}}(\theta_0^{(\mathrm{L}),r})\| \\
&\leq 2\eta G L\|\theta_k^{(\mathrm{L}),r} - \theta_0^{(\mathrm{L}),r}\| \\
&= 2\eta^2 G L\|\sum_{j=0}^{k-1}\nabla f_{\mathrm{L}}(\theta_j^{(\mathrm{L}),r})\| \\
&\leq 2\eta^2 L T G^2.
\end{aligned}
$$

Telescoping the inequality in equation 5 over $r = 1, ..., R$ and rearranging them, we obtain that

$$
\frac{1}{R}\sum_{r=0}^{R-1}\|\nabla f_{\mathrm{L,P}}(\theta^r)\|^2 \leq \frac{f_{\mathrm{L,P}}(\theta^0) - \inf_\theta f_{\mathrm{L,P}}(\theta)}{\eta R} + 3\eta L T G^2.
$$

Choosing $\eta = \sqrt{\frac{f_{\mathrm{L,P}}(\theta^0) - \inf_\theta f_{\mathrm{L,P}}(\theta)}{3RLTG^2}}$, we finally obtain that

$$
\min_{0\leq r\leq R-1}\|\nabla f_{\mathrm{L,P}}(\theta^r)\|^2 \leq \frac{1}{R}\sum_{r=0}^{R-1}\|\nabla f_{\mathrm{L,P}}(\theta^r)\|^2 \leq \sqrt{\frac{12LTG^2}{R}\left(f_{\mathrm{L,P}}(\theta^0) - \inf_\theta f_{\mathrm{L,P}}(\theta)\right)}.
$$

Consequently, $\min_{0\leq r\leq R-1}\|\nabla f_{\mathrm{L,P}}(\theta^r)\| \to 0$ as $R \to \infty$. This completes the proof. We note the above complexity bound may not be tight due to the two $\arg\min$ operations. We conjecture that the assisted gradient descent can achieve the same order of convergence rate as the gradient descent algorithm in nonconvex optimization.

**Remark**: If the learning rate $\eta$ is small, the Inequality (3) shows that the gradient $\nabla f_{\mathrm{L}}(\theta_k^{(\mathrm{L}),r})$ should not be well aligned with the gradient of the global loss $\nabla f_{\mathrm{L,P}}(\theta_k^{(\mathrm{L}),r})$. Intuitively, this is because $\theta_k^{(\mathrm{L}),r}$ is the best model chosen from the training trajectory $\{\theta_t^{(\mathrm{L}),r}\}_t$ that achieves the minimum global loss, and therefore the subsequent gradient update $\nabla f_{\mathrm{L}}(\theta_k^{(\mathrm{L}),r})$ must be badly correlated with $\nabla f_{\mathrm{L,P}}(\theta_k^{(\mathrm{L}),r})$ so that the global loss actually increases after the next gradient descent iteration.

## A.2 Visualization of AssistSGD Training: A Regression Example

We apply the AssistSGD to solve a regression problem with simulated data $a = [-1, -1]$, $b = [1, -1.25]$ and hyperparameters $T = 10$, $\eta = 0.9^r$ for both the learner and the provider. We run the algorithm for $R = 10$ assistance rounds. Figure 6 shows the learning trajectory of $\theta^r$ for $r = 0, 1, \ldots, 9$. It can be seen that at the beginning, the learner L's learning trajectory moves toward the oracle solution since the directions of two local optima are roughly the same; then, it oscillates in between two opposite directions and converges to the oracle solution.

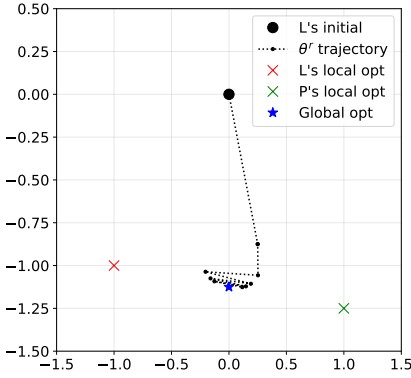

Figure 6: Learning trajectory of AssistSGD in regression.

## A.3    ADDITIONAL EXPERIMENTS OF ASSISTED DEEP LEARNING

### A.3.1    EFFECT OF LEARNER'S DATASIZE AND IMBALANCE LEVEL

**CIFAR-10 Dataset**

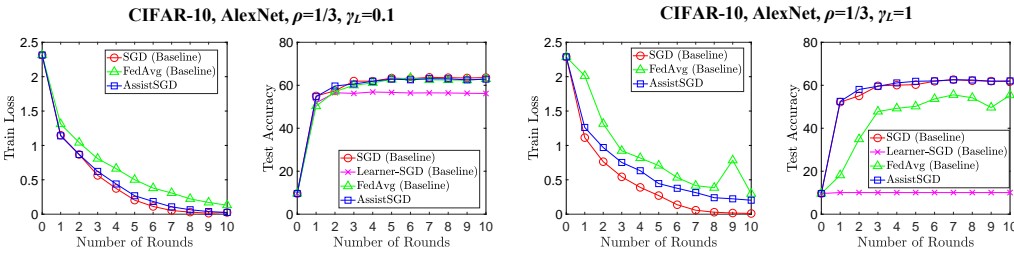

Figure 7: Comparison of AssistSGD, SGD, Learner-SGD and FedAvg with $\rho = 1/3$ using AlexNet.

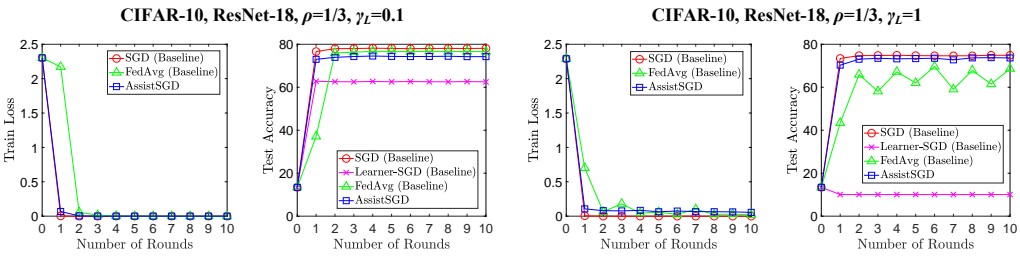

Figure 8: Comparison of AssistSGD, SGD, Learner-SGD and FedAvg with $\rho = 1/3$ using ResNet-18.

In this subsection, we present more CIFAR-10 experimental results of training AlexNet (Figure 7) and ResNet-18 (Figure 8) under the data size ratio $\rho = 1/3$ and different levels of data imbalance. We test and compare our AssistSGD with SGD, Learner-SGD, and FedAvg with balanced ($\gamma_L = 0.1$) and imbalanced ($\gamma_L = 1$) learner's data. From both figures, we can draw the same conclusions as those made in Section 4.2.

**SVHN Dataset**

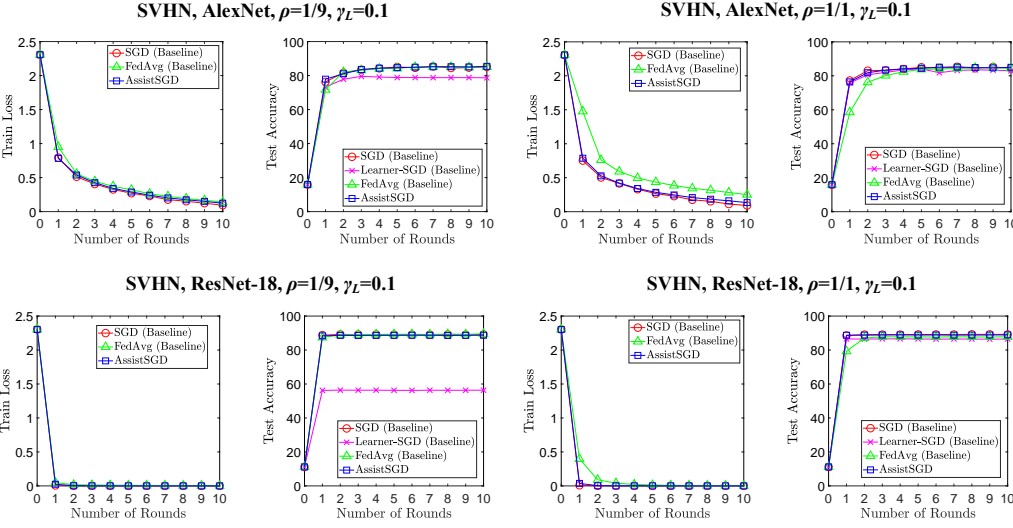

Figure 9: Comparison of AssistSGD, SGD, Learner-SGD and FedAvg on SVHN with balanced learner's data using AlexNet (top row) and ResNet-18 (bottom row).

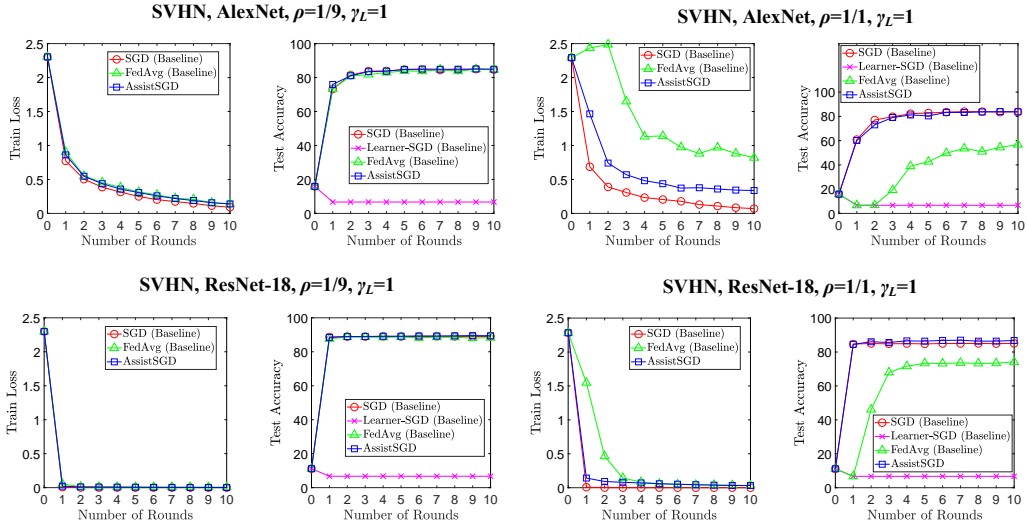

Figure 10: Comparison of AssistSGD, SGD, Learner-SGD and FedAvg on SVHN with imbalanced learner's data using AlexNet (top row) and ResNet-18 (bottom row).

In this subsection, we present the SVHN experimental results with balanced learner's data $\gamma_L = 0.1$ (Figure 9) and imbalanced learner's data $\gamma_L = 1$ (Figure 10) corresponding to the CIFAR-10 results in Section 4.2. From both figures, we can draw the same conclusions as those made in Section 4.2, which proves that our proposed AssistSGD can work well on a wide range of dataset types.

### A.3.2  EFFECT OF SAMPLING PERIOD

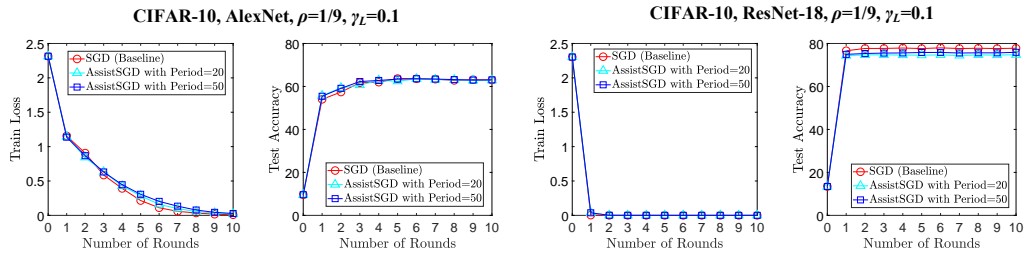

Figure 11: Comparison for AlexNet and ResNet-18 with balanced learner's data ($\gamma_L = 0.1$) under different sampling periods.

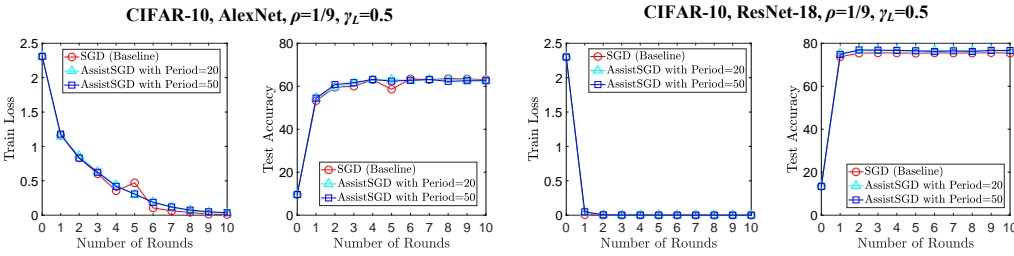

Figure 12: Comparison for AlexNet and ResNet-18 with imbalanced learner's data ($\gamma_L = 0.5$) under different sampling periods.

CIFAR-10, AlexNet, $\rho$=1/9, $\gamma_L$=1

CIFAR-10, ResNet-18, $\rho$=1/9, $\gamma_L$=1

Figure 13: Comparison for AlexNet and ResNet-18 with imbalanced learner's data ($\gamma_L = 1$) under different sampling periods.

We explore whether increasing the sampling frequency of the model (AlexNet and ResNet-18) and loss value can improve the performance of AssistSGD. Under data size ratio $\rho = 1/9$, we test and compare AssistSGD and SGD with balanced and imbalanced learner's data, i.e., $\gamma_L = 0.1, 0.5, 1$. We set the sampling period to be 20 and 50. The comparison results are shown in Figures 11, 12 and 13. It can be seen that using a low sampling frequency for AssistSGD already achieves the baseline performance of SGD. It implies that AssistSGD does not require much information exchange between the learner and provider. This helps save computation resources and reduce information leakage.

## A.4 Visualizations of the Reinforcement Learning Games

In this section, we visualize the landing trace of the LunarLander trained by AssistPG and Leaner-PG in different test environments.

Specifically, we consider a fixed map and set the engine power of the lander to be 10, 20, 30, and 40, respectively. In each setting, we train the lander using both AssistPG and Learner-PG for $R = 5$ rounds. After each round of training, we let the lander play an episode using the trained model and plot the corresponding landing trace. These traces are plotted in Figures 14, 15, 16, 17. From these figures, it can be seen that the lander with engine power 20-40 trained by AssistPG can successfully land to the landpad after 5 rounds of assisted learning. As a comparison, the lander trained by Learner-PG cannot even land after 5 rounds of training. This demonstrates the advantage of AssistPG. On the other hand, when the lander has a small engine power 10, it is challenging for both algorithms to land the lander properly, as the engine cannot provide sufficient acceleration.

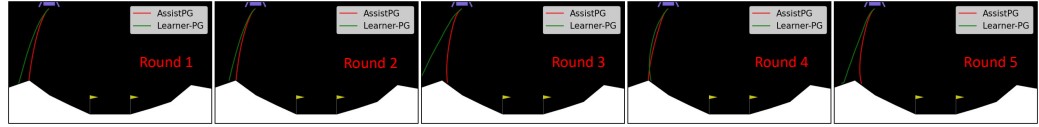

Figure 14: Comparison of landing traces of LunarLander with engine power = 10 trained by AssistPG and Learner-PG.

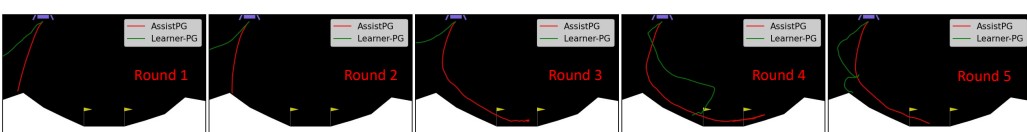

Figure 15: Comparison of landing traces of LunarLander with engine power = 20 trained by AssistPG and Learner-PG.

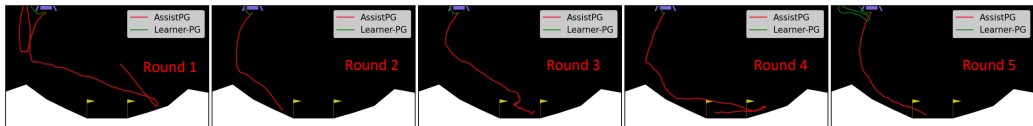

Figure 16: Comparison of landing traces of LunarLander with engine power = 30 trained by AssistPG and Learner-PG.

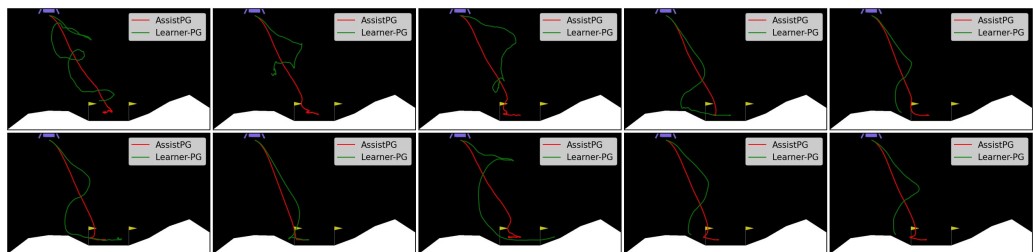

Figure 17: Comparison of landing traces of LunarLander with engine power = 40 trained by AssistPG and Learner-PG.

Moreover, after 5 rounds of training (using both AssistPG and Learner-PG), we test the lander in both the test environment I ("Test I") and II ("Test II"), and plot the landing traces in Figures 18 and 19, respectively. Here, for each test, we consider a fixed map and randomly generate 10 different engine powers from $\text{Uniform}(10, 40)$ (for Test I) and $30 * \text{Beta}(5, 1) + 10$ (for Test II).

From both figures, it can be seen that the lander trained by the AssisPG lands more smoothly in all test environments under diverse engine powers than that trained by the Learner-PG.

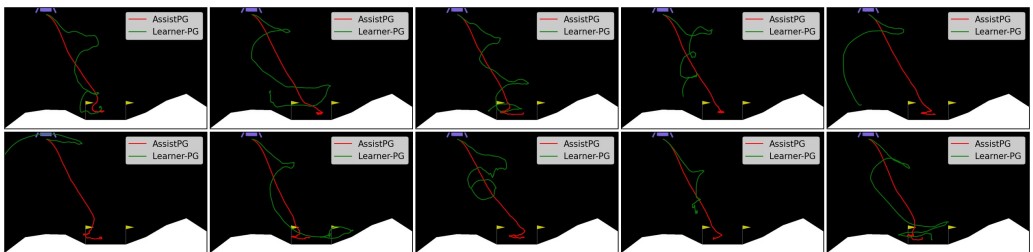

Figure 18: Comparison of landing traces of LunarLander with engine power $\sim$ $\text{Uniform}(10, 40)$ trained by AssistPG and Learner-PG.

Figure 19: Comparison of landing traces of LunarLander with engine power $\sim 30 * \text{Beta}(5, 1) + 10$ trained by AssistPG and Learner-PG.

The video version for the CartPole and LunarLander games can be accessed from the anonymous link `https://www.dropbox.com/sh/oz2jswj36li4lkh/AADaQn4Nj67v9mdIHKDLN6nAa?dl=0`. In the CartPole game, four videos record the performance of AssistPG and Learner-PG against the first five rounds with pole length equaling 1, 2, 3, and 4, respectively. Another two videos record 10 plays in the test environment I and II, respectively. In all the plays, both AssistPG and Learner-PG use the model trained from the fifth

round. In the LunarLander game, four videos record the performance of AssistPG and Learner-PG against the first five rounds with engine power equaling 10, 20, 30, and 40, respectively. Another two videos record 10 plays in the test environment I and II, respectively. In all the plays, both AssistPG and Learner-PG use the model trained from the fifth round. The videos show that with the assistance from the provider, the user can quickly generalize its model to more diverse environments.

