# OpenReview forum: "Assisted Learning for Organizations with Limited Imbalanced Data"
_ICLR.cc/2022/Conference — ICLR 2022 Submitted_

### Official Review · Reviewer_D8UY · 2021-11-03

**Correctness:** 4
**Technical Novelty And Significance:** 2
**Empirical Novelty And Significance:** 2
**Recommendation:** 5
**Confidence:** 3

**Main Review:**

- I think there may be a scenario assumed in the paper, but I'm curious to see if there are any specific scenarios in which the service provider already has enough data of the distribution that the learner wants to learn. Can you provide a more specific example? If there is an organization and service provider that will use the proposed framework in the current world, it would be good to give an example.

In experiments:
- It would be good to compare the performance when learning only with the data the provider has. I wonder whether the experimental settings are where learning is good enough only with the data the provider has or not.
- I understood that standard SGD (using centralized data) baseline train a model using centralized data.  Could you explain why the values change for each round in the graphs? What does "round" mean in this training?
- Could you elaborate on how you did federated learning in the baseline? I wonder how the action for each round was defined in federated learning and whether the comparison was made fair.


**Summary Of The Paper:**

This paper proposed a new assisted learning framework in which the service provider aims at helping the organization train a model with limited and imbalanced data.

**Summary Of The Review:**

The paper proposed a simple and straightforward assisted learning framework for a learner with limited and imbalanced data. It seems that the paper can be improved by presenting specific examples of scenarios that require the proposed framework and supplementing some parts in the experimental part.

---

> ### Author Response · Authors · 2021-11-22
> **Thank you for the time and comments. We address each comment below.**
>
> **Q1.** I think there may be a scenario assumed in the paper, but I'm curious to see if there are any specific scenarios in which the service provider already has enough data of the distribution that the learner wants to learn. Can you provide a more specific example? If there is an organization and service provider that will use the proposed framework in the current world, it would be good to give an example.
>
> **A:** Thank you for the helpful suggestion. There are many realistic scenarios in the real world that potentially apply our proposed framework. For example, the service provider can be a medical center that legally possesses healthcare data to improve treatment efficacy, while the learner can be a local small hospital or clinic. Then, the small hospitals or clinics can query a few rounds of assisted learning services from this medical center to help improve their services without sharing data.
>
> **Q2.** It would be good to compare the performance when learning only with the data the provider has. I wonder whether the experimental settings are where learning is good enough only with the data the provider has or not.
>
> **A:** We have compared the performance when learning only with the provider’s data. For supervised deep learning, we name this algorithm Provider-SGD (parallel to Learner-SGD that trains using only the learner’s data). And for reinforcement learning, we name this algorithm Provider-PG (parallel to Learner-PG that trains using only the learner’s data).
>
> We have implemented several supervised learning experiments of Provider-SGD on CIFAR-10 using AlexNet and ResNet-18. From the results, we find that in general, the Provider-SGD achieves a comparable or slightly worse test performance to that of AssistSGD/SGD, a comparable test performance to that of FedAvg, and better test performance to that of Learner-SGD (especially when the Learner has a smaller and heterogeneous local data). On the other hand, with the provider holding more data (i.e., smaller $\rho$), Provider-SGD achieves a better performance than the Learner-SGD. For example, when we train AlexNet with $\rho=1/9$ (provider holds 8 times more data than learner), the Provider-SGD (63%) performs comparable to AssistSGD (63.1%) and SGD (63%), and performs comparable to FedAvg (62.8%), but performs much better than Learner-SGD (10%).
>
> Also, we have implemented several reinforcement learning experiments of Provider-PG on three testing environments: the testing environments I and II as illustrated in the paper, and testing environment III that generates the pole length from Beta(5,1) with probability 0.2 and Uniform(0,5) with probability 0.8. Note that all testing distributions cover both the learner’s and the provider’s environments. We find that the Provider-PG performs as good as AssistPG on testing environment II (as this environment is very close to the provider's environment), but worse than AssistPG on the testing environment I and III. For your reference, we attach our preliminary results in the past few days in Table 1.
>
> Table 1: Test reward for the task 1 in Environment  I, II & III
>
> assistPG               	900               	820              	900
>
> learnerPG             	550               	500              	750
>
> Provider-PG         	670                	800             	190
>
> PG                        	910                	800             	850
>
> FedAvg                	800                	780              	600
>
> **Q3.** I understood that standard SGD (using centralized data) baseline train a model using centralized data. Could you explain why the values change for each round in the graphs? What does "round" mean in this training?
>
> **A:** Suppose AssistSGD consumes K data samples in an assistance round, then a round of standard SGD is defined by consuming K samples (so x-axis aligns the sample complexity). For example, in our deep learning experiments, if the total number of learner’s and provider’s local iterations is 2000 in each round, then for the standard SGD, an assistance round on the x-axis is interpreted as 2000 regular training iterations. Thus, a round in standard SGD is similar to an epoch in the common case. So, it is understandable that the values of standard SGD will change for each round.
>
> We interpreted standard SGD in this way in order to fairly compare with AssistSGD.
>
> === To Be Continued (more responses are appended in the following text field) ===

---

> > ### Author Response · Authors · 2021-11-22
> > **== Response Continued ==**
> >
> > **Q4.** Could you elaborate on how you did federated learning in the baseline? I wonder how the action for each round was defined in federated learning and whether the comparison was made fair.
> >
> > **A:** In federated learning (FedAvg in our experiments), each agent/learner performs multiple (1000 in Deep Learning experiments) local SGD steps, and sends the last model to the cloud. The cloud then makes the average of the model and sends it back to the local agents/learners to start the new local training. As described in Section 4.2, for the federated learning algorithm FedAvg, we treat the learner and provider as two federated learning agents, and they both inherit the same local data and local SGD iteration budgets from AssistSGD. In sum, we regard the two agents of federated learning as the learner and provider of assisted learning, and all settings of federated learning keep the same as those of assisted learning, except that the federated learning model trains following the process of FedAvg algorithm. The comparison is fair because FedAvg and AssistSGD have the same computation complexity at each round.

---

### Official Review · Reviewer_Bk5z · 2021-11-04

**Correctness:** 3
**Technical Novelty And Significance:** 3
**Empirical Novelty And Significance:** 3
**Recommendation:** 5
**Confidence:** 3

**Main Review:**

Strengths:
1. The problem setting is novel and may be relevant for real-life deployment of ML.
2. The theoretical analysis is rigorous.
3. The empirical evaluations test several different tasks and models.

Weaknesses:
1. Regarding the setup: Even though no data is exchanged, exchanging the models may leak data. Unbounded communication is unrealistic.
2. Regarding algorithm 1: How are the models in $\tau, \tau'$ selected? What's the relationship between the number of models selected and the convergence rate or accuracy?
3. Theorem 1 is an asymptotic result for $r \rightarrow \infty$, but the goal of the paper is to minimize the number of rounds, so concrete bounds on $R$ are desired, i.e. $R$ should appear in the bound.
4. No error bars in section 4.2.

**Summary Of The Paper:**

This paper studies a novel problem setup where a learner has unbalanced data, and a service provider has complementary or sufficient data, and the learner needs to improve accuracy in as few rounds of communication as possible, where the communication in each round is unbounded. An algorithm AssistSGD is proposed and shown to converge to a stationary point. Experiments show that AssistSGD has better performance than baselines and is close to centralized SGD on CIFAR 10 and reinforcement learning tasks,

**Summary Of The Review:**

The paper considers a novel problem setting and has a promising direction. The theoretical part can be enhanced.

---

> ### Author Response · Authors · 2021-11-22
> **Thank you for the time and comments. We address each comment below.**
>
> **Q1.** Regarding the setup: Even though no data is exchanged, exchanging the models may leak data. Unbounded communication is unrealistic.
>
> **A:** In our algorithm, we only exchange the models that are separated multiple iterations (30-50 iterations) apart. In this way, it is difficult to infer data information. As for the unbounded communication, we made this ideal assumption to make our theoretical analysis and empirical experiments concise and convenient to prove the validity of our proposed algorithms. A more realistic situation is that the communication is very large and enough to implement the proposed assisted learning, which won’t change the results.
>
> **Q2.** Regarding algorithm 1: How are the models in $\tau$, $\tau$' selected? What's the relationship between the number of models selected and the convergence rate or accuracy?
>
> **A:** As we mentioned in the Discussions of Section 2.2, in each local SGD training of Algorithm1, we sample the training iterations at a low frequency (i.e., 50 in our experiments) which means we pick the model out every 50 iterations. The sampled iterations sets are denoted as $\tau$ and $\tau$’.
>
> As for the relationship between the number of models selected and the convergence rate or accuracy, please refer to the empirical results of the “effect of sampling period” (Section A.3.2) in the Appendix. Theoretically, the more models we select, the faster the convergence rate and the better the accuracy, but it will cost many computation resources. However, from the results, using a low sampling frequency for AssistSGD already achieves the baseline performance of SGD, which implies that AssistSGD does not require much information exchange between the learner and provider. This helps save computation resources and reduce information leakage.
>
> **Q3.** Theorem 1 is an asymptotic result for $R \to \infty$, but the goal of the paper is to minimize the number of rounds, so concrete bounds on $R$ are desired, i.e. $R$ should appear in the bound.
>
> **A:** We showed by experiments that only a few (e.g., ten) assistance rounds are often enough to obtain near-convergence and satisfactory results. Our current theoretical analysis only shows convergence for general learning objectives, with a loose upper bound of the rate of convergence.
>
> **Q4.** No error bars in section 4.2.
>
> **A:** Thanks for the suggestion, we will report the error bar in the revised version.

---

### Official Review · Reviewer_cvfa · 2021-11-09

**Correctness:** 2
**Technical Novelty And Significance:** 2
**Empirical Novelty And Significance:** 2
**Recommendation:** 3
**Confidence:** 4

**Main Review:**

Below please find a few questions/ concerns on the paper, which I hope the authors can clarify/address:
* At a motivational level, privacy is an important consideration that motivates the problem, but discussion on data privacy of the proposed protocol is missing. When the learner sends the model parameters along with the local losses to the service provider, there is no guarantee that data privacy is still preserved. Can the provider infer the learner’s data distribution from the exchanged information?

* In the 2nd step of the AssistSGD and AssistPG algorithms, the service provider outputs the best model among the ones received from the learner with the minimal global loss; in the 3rd step of the AssistSGD and AssistPG algorithms, the learner outputs the best model among the ones received from the service provider with the minimal *global* loss. How can the learner evaluate the global loss D^{(L,P)} without access to D^{(P)} (i.e. with no data sharing as specified in the assisted learning protocols)? Is there an assumption that both the learner and the provider share a validation dataset based on which they select the models  θ^{(P)}_0 and θ^r?

* The consistency result is rather weak considering that a key objective in the desiderata is to achieve limited assistance.

* For the assisted deep learning experiments, the service provider is set to have access to the (unbiased) underlying data distribution (i.e. data sampled uniformly at random from each class of CIFAR-10 and SVHN); this is a rather limited scenario as the service provider should already (intuitively) return a model with decent global loss.

* The paper claims that the proposed assisted learning framework is developed to facilitate the deployment of general machine learning in large organizations. However, the empirical sections do not cover any realistic use cases under the assisted learning protocol in the proposed context.


**Summary Of The Paper:**

This paper investigates a novel learning scenario, where the learner has limited access to the global data distribution and can share learned model parameters with a so-called service provider through multiple (but limited) rounds of interactions. The motivation is interesting and seemingly useful for the scenarios described in the introduction. The authors propose an intuitive assisted learning framework applicable to both (deep) supervised learning and reinforcement learning, and experiments show that the proposed algorithms achieve comparable performance with learning from centralized data. However, there are a few concerns/ questions in the problem setup and the proposed assisted learning protocol, please see the detailed review below.

**Summary Of The Review:**

The paper investigates an interesting learning protocol, proposes an intuitive assisted learning framework, but lacks sufficient theoretical justification and empirical support for the significance of the solution.

---

> ### Author Response · Authors · 2021-11-22
> **Thank you for the time and comments. We address each comment below.**
>
> **Q1.** At a motivational level, privacy is an important consideration that motivates the problem, but discussion on data privacy of the proposed protocol is missing. When the learner sends the model parameters along with the local losses to the service provider, there is no guarantee that data privacy is still preserved. Can the provider infer the learner’s data distribution from the exchanged information?
>
> **A:** We note that although the focus is not the quantification of data privacy, it is straightforward to enhance data privacy (e.g., in the sense of differential privacy) by injecting noises. One way is to replace the local SGD training of both the learner and provider with the differentially private SGD proposed in (Abadi et al., 2016). We did not include that experiment because it is not quite related to the focus of this work.
>
> On the other hand, at a motivation-level, our focus (as discussed in the paper) is to allow learners to assist each other without sharing data and within a small number of intersections. The setup is similar to conventional distributed learning in terms of data forms and objective functions, but different in the following aspects. First, the communication bandwidth is not a concern for organizational learners. Instead, our approach aims to reduce that communication rounds since it is often the bottleneck of organizational learners (e.g., hospitals, government agencies, in collaborative learning). Second, our approach is particularly suitable for imbalance and heterogeneous data, significantly outperforming the state-of-the-art FedAvg method (as shown in experiments).
>
> **Q2.** In the 2nd step of the AssistSGD and AssistPG algorithms, the service provider outputs the best model among the ones received from the learner with the minimal global loss; in the 3rd step of the AssistSGD and AssistPG algorithms, the learner outputs the best model among the ones received from the service provider with the minimal global loss. How can the learner evaluate the global loss D^{(L,P)} without access to D^{(P)} (i.e. with no data sharing as specified in the assisted learning protocols)? Is there an assumption that both the learner and the provider share a validation dataset based on which they select the models θ^{(P)}_0 and θ^r?
>
> **A:** As we described in Section 2.1, the global loss can be evaluated by summing up the separated local losses, namely f(theta; D^(L, P)) = f(theta; D^(L)) + f(theta; D^(P)). Thus, after receiving the local loss f(theta; D^(P)) from the provider P, the learner L can evaluate the global loss by computing the local loss f(theta; D^(L)) and summing it with f(theta; D^(P)). The same processing also works for provider P. Note that both the model parameters and local loss will be delivered between the learner L and provider P.
>
> **Q3.** The consistency result is rather weak considering that a key objective in the desiderata is to achieve limited assistance.
>
> **A:** We showed by experiments that ten assistance rounds are often enough for convergence and satisfactory results. Our theoretical analysis shows convergence for quite general learning objectives and an explicit upper bound of rate of convergence. With more assumptions on learning objectives and data assumptions, one may further tighten the bounds.
>
> **Q4.** For the assisted deep learning experiments, the service provider is set to have access to the (unbiased) underlying data distribution (i.e. data sampled uniformly at random from each class of CIFAR-10 and SVHN); this is a rather limited scenario as the service provider should already (intuitively) return a model with decent global loss.
>
> **A:** We performed several scenarios considering different provider’s distributions. Even in the case that both learner and provider have balanced data, the provider still cannot return a model with a decent global loss, because the data size is limited. As shown on the right-hand side of Figure 2, learner and provider hold balanced data, and they have identical data sizes, the provider’s performance should be similar to the learner-SGD. We can see from the results that the AssistSGD still outperforms learner-SGD, which means it outperforms provider-SGD as well.
>
> **Q5.** The paper claims that the proposed assisted learning framework is developed to facilitate the deployment of general machine learning in large organizations. However, the empirical sections do not cover any realistic use cases under the assisted learning protocol in the proposed context.
>
> **A:** The main purpose of the paper is to provide an assisted learning protocol for organizational-level learners; those learners have abundant computation sources, but they are restricted to limited communications. As we explained in the paper, organizational-level learners may not make data public. Thus, we could only use some popular and well-studied benchmark image data and reinforcement learning tasks to implement the experiments.

---

### Author Response · Authors · 2021-11-22
**We thank all the reviewers for their time and constructive comments.**

We have successfully addressed all the comments below. All the related discussions will be incorporated into the revised paper.

---

### Decision · Program_Chairs · 2022-01-20

**Decision:**

Reject

**Comment:**

The paper proposed a novel assisted learning scenario which would likely be useful for organizational level learners (i.e. learners with sufficient computational resources but limited and imbalance data). The paper is generally well presented, but there are shared concerns amongst the reviewers in the significance of technical contributions: (1) Due to the asymptotic nature of the consistency results, the technical strength is not strongly supported with the existing theoretical analysis. (2) Although the problem setup is novel and seems interesting, the practical significance of the results is not well supported without a concrete real-world application. (3) There are a few clarity issues raised in the reviews, which suggest that the paper could benefit from a major revision to address the above concerns.